# The mediating role of internal motivation on the relationship between ethical leadership and employee performance in hospitals in Northern Jordan

Raya Al-Bataineh[1]*, Ameera Hayajneh[2]

**1** Faculty of Medicine, Department of Health Management and Policy, Jordan University of Science and Technology, Irbid, Jordan, **2** Faculty of Medicine, Department of Health Management and Policy, Jordan University of Science and Technology, Irbid, Jordan

* rtalbataineh0@just.edu.jo

## Abstract

### Objective

The study's main aim is to investigate the influence of ethical leadership (EL) on employee performance (EP) through internal motivation (IM) from the perspectives of clinical and administrative employees working in hospitals in northern Jordan.

### Method: Design

The study used a descriptive, correlational cross-sectional quantitative design.

### Participants and setting

Data were collected from 330 clinical and administrative employees between February and March 2024 using convenience sampling from five hospitals—2 public, 2 private, and 1 teaching hospital—in different geographical areas in northern Jordan. The study hypotheses were tested using a hierarchical multiple linear regression.

### Results

The study results revealed a statistically significant association between ethical leadership, internal motivation, and employee performance. Moreover, the results showed that internal motivation statistically mediates the relationship between ethical leadership and employee performance.

### Conclusion

The current study's findings can serve as an empirical basis for hospital decision-makers to plan and implement programs and/or establish or revise policies for the target population, improving employees' performance, achieving desired outcomes, and ultimately providing better care for patients.

**Data availability statement:** All relevant data are within the manuscript and its Supporting information files.

**Funding:** This study was funded by Jordan University of Science and Technology (https://www.just.edu.jo/Pages/Default.aspx) with the grant number of [ 2023/579]. The fund was received by Hahajneh, A. The funder had no role in the study design, data collection and analysis, decision to publish, or preparation of the manuscript.

**Competing interests:** The authors have declared that no competing interests exist.

# 1. Introduction

Leadership is fundamental to organizational success, as demonstrated by the fact that it has been a research subject for over fifty years [1]. Leadership is defined as an individual's action while guiding a group's activities toward a common objective [2,3]. Ethical leadership is one of the most important topics that attracts the attention of modern organizations, as it forms the core ethical values and principles upon which leaders build their success and positive influence [4,5]. Brown and his colleagues defined ethical leadership as exhibiting ethically suitable behavior through personal and interpersonal relationships and promoting such behavior to followers through two-way communication, reinforcement, and decision-making [6]. This is done through ethical leaders who are driven by moral beliefs, caring values, and ethics [7].

Ethical leaders possess qualities such as ethical sight, participating in ethical decision-making, and prioritizing long-term decisions [8]. They treat employees fairly, set an example of ethical behavior, and effectively communicate the value of ethics to employees. These qualities promote the development of knowledge of appropriate behavior among employees as well as their learning and ability to model the actions of leaders, all of which improve how well they perform at work [9]. According to [10], employee performance essentially refers to the accomplishments and results accomplished at work. Also, it is the ability to remain committed to a plan and achieve the desired outcomes. To attain high performance, leaders must create a work environment conducive to productivity for their organizations. This is done by motivating employees who have a vital and effective role to obtain high performance [11]. Sihag defined motivation as an art that involves setting up situations that allow people to work at their highest productivity levels while also receiving compensation, benefits, and encouragement [12].

## 1.2. Literature review

Studies from various sectors highlights the critical role of ethical leadership in enhancing employees' performance, well-being, and organizational outcomes, especially in healthcare settings. In China, [13], ethical leadership positively influenced nurses' self-compassion and job performance, with 27% of the overall effect was partially mediated by self-compassion. Similarly, in Austria [14], it was associated with lower burnout, higher job satisfaction and affective commitment, demonstrating its role in fostering supportive work environments.

A systematic review of 11 studies identified six domains -autonomy, competencies, relatedness, individual nursing characteristics, relationships and support, and leadership practice- through which nurse leaders' behaviors directly and indirectly influence staff performance [15]. Ethical leadership has been found to positively influence nurses' organizational citizenship behavior (OCB) by strengthening trust and psychological well-being [16], promote nurse well-being with workplace mindfulness partially mediating the effect [17], and increase whistleblowing intentions through psychological safety [18]. Similarly ethical leadership and employee flourishing predict extra-role behavior, reinforcing the idea that ethical leaders encourage employees to go beyond their formal job duties [19,20].

Beyond healthcare, studies consistently link between ethical leadership and internal motivation and positive workplace outcomes. Research in tourism, public organization, education, and banking [21–24] shows that ethical leadership boosts internal motivation, which in turn, enhance performance, innovation, and organizational commitment. Evidence from Greece, Jordan, China, and the UAE [25–28], together confirms its cross cultural effectiveness in fostering motivation, satisfaction, and performance.

In parallel, ethical leadership has been widely recognized as a driver of employees' prosocial behaviors, work outcomes, and organizational effectiveness. It enhances helping behaviors and (OCB) through trust and organizational justice [29], positively influenced job performance, satisfaction, self-efficacy, and employee voice [30] and reduced turnover intentions by building trust and commitment [31]. From a signaling theory perspective, ethical leaders signal support and integrity, thereby fostering role breadth self-efficacy and prosocial motivation, which enhance innovative work behaviors [32].

Collectively, this growing body of evidence suggests that ethical leadership contributes significantly to employee performance, well-being, prosocial behavior, and organizational effectiveness, both directly and indirectly through mediators such as self-compassion, mindfulness, trust, psychological safety, internal motivation, organizational identification, and prosocial motivation. In Jordan, hospitals including public, private, and university hospitals face high stress and resource constraints, intensified by the influx of refugees, making leadership crucial [33–35]. Ethical leadership characterized by fairness, integrity, and support fosters trust, morale, and performance [36] with internal motivation acting as a critical mechanism linking ethical leadership to employee performance [37]. Despite its potential, the extent to which ethical leadership impacts performance in Jordanian hospitals is under-researched [38]. This study seeks to fill the gap by investigating the relationship between ethical leadership, internal motivation, and employee performance in hospitals in northern Jordan, providing evidence to inform leadership practices in challenging healthcare settings.

Drawing on the literature, the study's proposed four hypotheses as following:

| Hypothesis | Statement |
|---|---|
| $H_1$ | Ethical leadership has a significant positive association with internal motivation among clinical and administrative employees in northern Jordanian hospitals (public/private/university) |
| $H_2$ | Internal motivation has a significant positive association with employee performance among clinical and administrative employees in northern Jordanian hospitals (public/private/university) |
| $H_3$ | Ethical leadership has a significant positive association with employee performance among clinical and administrative employees in northern Jordanian hospitals (public/private/university). |
| $H_4$ | Internal motivation has a significant positive impact on the relationship between ethical leadership and employee performance among clinical and administrative employees in northern Jordanian hospitals (public/private/university). |

## 2. Methods

### 2.1. Design, sample, and settings

The study was conducted using a descriptive, correlational, cross-sectional quantitative design. Convenience sampling was used to recruit the participants who were clinical and administrative employees working in five hospitals (2 private hospitals, 2 public hospitals, 1 university hospital) located in northern Jordan. Criteria for participation in the study included (1) having an administrative and/or clinical role, (2) currently working in the study setting, including the five hospitals, (3) being able to participate in the study voluntarily, and (4) had a minimum of one year of work experience. Employees who were on leave, had temporary assignments, and working experience less than one year were excluded from the study.

### 2.2. Sample size

The sample size of the current study was calculated by using G*Power. Assuming α of 0.05, a power of 0.95 (the generally required largest sample size), an effect size of 0.075, and a maximum of 9 tested predictors (the main study independent variables and demographic variables). The resulting required sample size was 330.

## 2.3. Instruments

The study data was collected using an Arabic-structured questionnaire consisting of closed-ended questions. The questionnaire consisted of four parts including, employee characteristics, the Ethical Leadership Scale, the Internal Motivation Scale, and the Employee Performance Scale.

**2.3.1**. **The employee characteristics.** The first section was devoted to the employee characteristics of the study sample, including age, gender, marital status, educational level, hospital type, years of experience, and job position/ profession.

**2.3.2**. **The ethical leadership scale.** The second section contained ten items to measure the independent variable represented by ethical leadership. These items were borrowed from Yukl and his colleagues [39]. The items were on a 6-point Likert-type scale, ranging from (1 = strongly disagree, 6 = strongly agree). The total score ranged between 10–60 with higher scores indicated stronger ethical leadership. Sample item include: 'My supervisor keeps his or her actions consistent with his or her stated values (walks the talk)'. The category interval was calculated as $((6-1) \div 3) = 1.67$. Accordingly, mean scores were interpreted as Low (1.00–2.67), Medium (2.68–4.32), and High (4.33–6.00). Cronbach's alpha values in these studies were above 0.9 [40].

**2.3.3**. **The internal motivation scale.** The third section included six items to measure the mediator variable represented by internal motivation, according to Abu Yahya (2018). The items were on a 5-point Likert-type scale ranging from 1 = strongly disagree to 5 = strongly agree. Sample items include: 'When choosing jobs, I usually choose the one that sounds like the most fun'. The total score ranged between 6–30 with higher scores indicated greater internal motivation. The category interval was calculated as $((5-1) \div 3) = 1.33$. Accordingly, mean scores were interpreted as Low (1.00–2.33), Medium (2.34–3.66), and High (3.67–5.00). Cronbach's alpha for the scale was 0.811 [40].

**2.3.4**. **The employee performance scale.** In the fourth section, the 20-item scale was used to measure the dependent variable represented by employee performance, according to Williams and Anderson, 1991. The items were on a 5-point Likert-type scale ranging from 1 = strongly disagree to 5 = strongly agree. The total score ranged between 20–100 with higher scores indicated higher performance. Sample items include: 'I adequately complete assigned duties,' 'I help others who have been absent'. The category interval was calculated as $((5-1) \div 3) = 1.33$. Accordingly, mean scores were interpreted as Low (1.00–2.33), Medium (2.34–3.66), and High (3.67–5.00). Cronbach's alpha values in these studies were above 0.7 [33].

A pilot study with 32 participants was conducted to test the study's instruments. Given the high reliability (Cronbach's $\alpha = 0.86$) and positive feedback, no modifications were needed, and these participants were excluded from the final sample.

**2.3.5**. **Validity and reliability of the study instrument.** Cronbach's alpha was used to assess internal consistency, with values of 0.70 or higher considered acceptable. The results indicate that the instrument is reliable: ethical leadership ($\alpha = 0.949$), internal motivation ($\alpha = 0.775$), employee performance ($\alpha = 0.790$), and overall instrument ($\alpha = 0.874$). Validity was ensured through adoption from established studies (Yukl et al., 2013; Potipiroon & Ford, 2017; Abu Yahya, 2018; Williams & Anderson, 1991; López-Cabarcos et al., 2022) and by conducting face validity checks with qualified academics, including verification during Arabic translation.

## 2.4. Data collection procedure

An approval from the Institutional Review Board (IRB) at the Jordan University of Science and Technology with the reference number (IRB:2023/579) and permissions from the study settings were obtained prior to data collection. The researchers conducted the data collection process during February and March of 2024. Data collection was conducted in person using a physical paper-based questionnaire. Human resources departments in the included hospitals were contacted to obtain permission and facilitate access for 330 participants. In each of the five hospitals, the data collection process was facilitated by a designated staff member assigned by the hospital administration to support participant

outreach and engagement. Employees who met the study's inclusion criteria were asked to participate and then given the questionnaire to fill out in a private place. A written consent form was obtained from the participants before filling out the questionnaires. The participants were allocated approximately 8–10 minutes to complete the questionnaire. The participation in the study was entirely voluntary, and participants had the freedom to withdraw from the study at any point without any repercussions. The participants received assurances of the confidentiality of the information collected. No harm was caused to the participants. In addition, no personal information revealing the participants' identities was included in the questionnaire.

### 2.5. Data analysis

Statistical analysis was conducted using SPSS software (version 25). Descriptive statistics was used to describe the study sample and variables. The study hypotheses were tested using a hierarchical multiple linear regression according to Baron and Kenny, 1968. Prior to starting the hypotheses analyses, the assumptions of the linear regression analysis including normality and absence of multicollinearity were examined for violations. Normality was assessed using the Kolmogorov-Smirnov test and Q-Q plots while the Variance Inflation Factor (VIF) test and Tolerance test were used to assess the absence of multicollinearity.

## 3. Results

### 3.1. Linear regression assumptions

**3.1.1**. **Normality.** Normality was tested, and the results indicated that the data was normally distributed, as evidenced by One-Sample Kolmogorov-Smirnov test with a p-value of bigger than (0.05) [35]. Q–Q plots further confirmed this, showing data points generally aligned with the diagonal line, with only minor deviations (see the supplementary file S1 File).

**3.1.2**. **Multicollinearity.** Multicollinearity was tested, and the results are presented in supplementary file S2 and S3 Files., assessing the Variance Inflation Factors (VIF) and tolerance values. The findings indicate absence of multicollinearity, with VIF values within the acceptable statistical range of 1.00. Furthermore, the tolerance values were 1.00, surpassing the threshold of 0.05 [36].

### 3.2. Employee demographic characteristics

The demographic characteristics of the study sample are presented in S.F.3. A total of 330 questionnaires were filled out by clinical and administrative employees. The majority of the participants were clinical employees, married, female, and fall in the 25–35 years age group. Over half of the participants have a bachelor's degree. Participants are distributed across university (27.3%), public (39.4%), and private organizations (33.3%) with the largest group having 11–20 years of experience.

### 3.3. Description of ethical leadership, internal motivation, and employee performance

The basic descriptive statistics of ethical leadership, internal motivation, and employee performance are shown in Table 1. With a mean score of 4.55 and a standard deviation of 0.90, the variable "Ethical Leadership" had the highest mean score. With a mean score of 3.80 and a standard deviation of 0.57, the variable known as "Employee Performance" had the lowest mean score. A detailed table of the descriptive statistics for each item of the study variables is provided in the supplementary file S4 and S5 Files.

Prior to hypotheses testing, multiple linear regressions were conducted to examine the associations between employees' demographic characteristics and the study's main variables. This step was undertaken to identify demographic factors that could potentially confound the relationships under investigation and should therefore be statistically controlled for in subsequent hierarchical regression analyses.

**Table 1. Descriptive Analysis (Mean and Standard Deviation) of Study Variables.**

| Variable Name | Mean | Standard Deviation |
|---|---|---|
| Ethical Leadership | 4.55 | 0.90 |
| Internal Motivation | 3.89 | 0.78 |
| Employee Performance | 3.80 | 0.57 |

**3.4.1. The employee demographic characteristics and internal motivation.** As shown in Table 2, only age and organization ownership were significantly associated with internal motivation.

**3.4.2. The employee characteristics and employee performance.** As shown in Table 3, educational level emerged as the only significant factor associated with employee performance ($p > 0.05$).

## 3.5. Testing the study hypotheses

Baron and Kenny (1968) proposed a four-step approach for testing the mediation by several regression analyses. In this study, the mediating variable was internal motivation, the independent variable was ethical leadership, and the dependent

**Table 2. The Calculations of Coefficients of Employee Characteristics and Internal Motivation.**

**Coefficients**

| Model | | Unstandardized Coefficients | | Standardized Coefficients | t | Sig. |
|---|---|---|---|---|---|---|
| | | B | Std. Error | Beta | | |
| 1 | (Constant) | 4.326 | .366 | | 11.829 | .000 |
| | Age** | −.200 | .076 | −.245 | −2.638 | .009 |
| | Gender | .008 | .096 | .005 | .083 | .934 |
| | Marital Status | .025 | .097 | .016 | .260 | .795 |
| | Education level | −.132 | .068 | −.110 | −1.942 | .053 |
| | Organization Ownership** | −.172 | .061 | −.171 | −2.839 | .005 |
| | Work experience | .080 | .051 | .147 | 1.561 | .120 |
| | Job category | .167 | .093 | .107 | 1.794 | .074 |

Note: Predictors include Age, Gender, Marital Status, Education Level, Organization Ownership, Work Experience, and Job Category. Outcome variable: Internal Motivation.

**Table 3. The Calculations of Coefficients of Employee Characteristics and Employee Performance.**

**Coefficients**

| Model | | Unstandardized Coefficients | | Standardized Coefficients | t | Sig. |
|---|---|---|---|---|---|---|
| | | B | Std. Error | Beta | | |
| 1 | (Constant) | 3.894 | .271 | | 14.393 | .000 |
| | Age | −.030 | .056 | −.050 | −.530 | .597 |
| | Gender | .052 | .071 | .042 | .728 | .467 |
| | Marital Status | .037 | .071 | .033 | .513 | .609 |
| | Education level** | −.108 | .050 | −.124 | −2.148 | .032 |
| | Organization Ownership | −.016 | .045 | −.021 | −.348 | .728 |
| | Work experience | .000 | .038 | −.001 | −.006 | .995 |
| | Job category | .044 | .069 | .039 | .638 | .524 |

Note: Predictors include Age, Gender, Marital Status, Education Level, Organization Ownership, Work Experience, and Job Category. Outcome variable: Employee Performance.

variable was employee performance. Table 3 presents the mediating influence of internal motivation on the relationship between ethical leadership and employee performance.

In the first step, a hierarchical multiple regression analysis with ethical leadership (X) predicting internal motivation (M) was conducted. Here the path is called direct effect. Given the potential influence of organization ownership and age on internal motivation, they were controlled by incorporating it into the first block of the equation in a hierarchical multiple regression analysis. Ethical leadership was entered into the second block.

In the second step, a hierarchical multiple regression with internal motivation (M) predicting employee performance (Y) was conducted. Here the path is called the direct effect. Given the potential influence of educational level on employee performance, it was controlled by incorporating it into the first block of the equation in a hierarchical multiple regression analysis. Internal motivation was entered into the second block.

In the third step, a hierarchical multiple regression with ethical leadership (X) predicting employee performance (Y) was conducted. Given the potential influence of educational level on employee performance, it was controlled by incorporating it into the first block of the equation in a hierarchical multiple regression analysis. Ethical leadership was entered into the second block.

Since significant relationships were found in steps 1–3, step 4 was subsequently conducted. In this step, a hierarchical multiple regression was conducted with ethical leadership (X) and internal motivation (M) predicting employee performance (Y). Here the path is called indirect effect. Given the potential influence of educational level on employee performance, it was controlled by incorporating it into the first block of the equation in a hierarchical multiple regression analysis. Internal motivation and ethical leadership were entered into the second block.

Since the effect of internal motivation on employee performance was still significant after controlling for ethical leadership and both ethical leadership and internal motivation significantly predicted employee performance, the findings support partial mediation. Also, after considering the effect of internal motivation, the impact of ethical leadership on employee performance declined but remained positively significant. In conclusion, the analysis showed that internal motivation partially mediates the association between ethical leadership and employee performance.

### 3.5.1. The first hypothesis of the study.

A hierarchical multiple regression analysis was employed to confirm the influence of ethical leadership on internal motivation, controlling for organization ownership and age. As shown in Table, ethical leadership significantly predicted internal motivation ($\beta = .522$, $p < 0.05$), indicating that the impact of ethical leadership is consistent across different employee groups. The overall model explained 22.2% of the variance in internal motivation ($R^2 = 0.222$, Adjusted $R^2 = 0.215$), and the effect size of adding ethical leadership was moderate ($f^2 \approx 0.286$), indicating meaningful practical significance in addition to statistical significance (Table 4).

**Table 4. The Calculations of Coefficients of The Hierarchical Multiple Regression of Ethical Leadership and Internal Motivation.**

| Model | | Unstandardized Coefficients | | Standardized Coefficients | t | Sig. | R² | Adjusted R² | f² |
|---|---|---|---|---|---|---|---|---|---|
| | | B | Std. Error | Beta | | | | | |
| 1 | (Constant) | 4.255 | .197 | | 21.604 | .000 | | | |
| | Organization ownership | −.138 | .056 | −.168 | −2.927 | .004 | 0.026 | 0.020 | 0.027 |
| | Age | .104 | .086 | −.066 | −1.141 | .225 | | | |
| 2 | (constant) | 3.001 | .224 | | 13.400 | .000 | | | |
| | Organization ownership | −.455 | .061 | −.453 | −7.514 | .000 | 0.222 | 0.215 | 0.286 |
| | Age | −.084 | .042 | −.103 | −1.988 | .048 | | | |
| | Ethical Leadership | .454 | .050 | .522 | 9.071 | .000 | | | |

Note: Predictors: Organization Ownership, Age, Ethical Leadership; Outcome: Internal Motivation.

**3.5.2. The second hypothesis of the study.** To examine the hypothesis, a hierarchical multiple regression analysis was employed to confirm the influence of internal motivation on employee performance, controlling for educational level. As shown in Table 5, internal motivation significantly predicted employee performance ($\beta = .479$, $p < 0.05$), indicating that employees with higher levels of internal motivation tend to exhibit better performance, regardless of demographic differences. The overall model explained 24.2% of the variance in employee performance ($R^2 = 0.242$, Adjusted $R^2 = 0.237$), and the effect size of adding internal motivation was large ($f^2 \approx 0.301$), indicating meaningful practical significance in addition to statistical significance.

**3.5.3. The third hypothesis of the study.** To examine the hypothesis, a hierarchical multiple regression analysis was employed to confirm the influence of ethical leadership on employee performance, controlling for educational level. As shown in Table 6, ethical leadership significantly influences employee performance ($\beta = .436$, $p < 0.05$). The model explained 19.9% of the variance in employee performance ($R^2 = 0.199$, Adjusted $R^2 = 0.194$), and the effect size of adding ethical leadership was moderate ($f^2 \approx 0.186$), indicating meaningful practical significance in addition to statistical significance.

**3.5.4. The fourth hypothesis of the study.** To examine the fourth hypothesis, a hierarchical multiple regression analysis was employed to confirm the influence of internal motivation on the relationship between ethical leadership and employee performance. Table 7 clearly shows that both internal motivation ($\beta = 0.386$, $p < 0.001$) and ethical leadership ($\beta = 0.323$, $p < 0.001$) significantly predicted employee performance. The full model explained 33.5% of the variance in employee performance ($R^2 = 0.335$, Adjusted $R^2 = 0.329$), with a large effect size for the added predictors ($f^2 \approx 0.322$), indicating meaningful practical significance. The effect of ethical leadership declined from $\beta = 0.436$ in the previous model (Table 6) to $\beta = 0.323$ after including internal motivation, suggesting that internal motivation partially mediates

**Table 5. The Calculations of Coefficients of The Hierarchical Multiple Regression of Internal Motivation and Employee Performance.**

| Model | | Unstandardized Coefficients | | Standardized Coefficients | t | Sig. | $R^2$ | Adjusted $R^2$ | $f^2$ |
|---|---|---|---|---|---|---|---|---|---|
| | | B | Std. Error | Beta | | | | | |
| 1 | (Constant) | 3.987 | .094 | | 42.396 | .000 | | | |
| | Educational Level | −.101 | .048 | −.115 | −2.104 | .036 | 0.013 | 0.010 | 0.013 |
| 2 | (Constant) | 2.632 | .160 | | 16.494 | .000 | | | |
| | Educational Level | −.077 | .042 | −.088 | −1.823 | .069 | 0.242 | 0.237 | 0.301 |
| | Internal Motivation | .348 | .035 | .479 | 9.924 | .000 | | | |

Note: Predictors in Step 1: Educational Level; Step 2: Educational Level and Internal Motivation. Outcome variable: Employee Performance.

**Table 6. The Calculations of Coefficients of The Hierarchical Multiple Regression of Ethical Leadership and Employee Performance.**

| Model | | Unstandardized Coefficients | | Standardized Coefficients | t | Sig. | $R^2$ | Adjusted $R^2$ | $f^2$ |
|---|---|---|---|---|---|---|---|---|---|
| | | B | Std. Error | Beta | | | | | |
| 1 | (constant) | 3.987 | .094 | | 42.396 | .000 | | | |
| | Educational Level | −.101 | .048 | −.115 | −2.104 | .036 | 0.013 | 0.010 | 0.013 |
| 2 | (constant) | 2.715 | .169 | | 16.084 | .000 | | | |
| | Educational Level | −.043 | .044 | −.050 | −.990 | .323 | 0.199 | 0.194 | 0.186 |
| | Ethical Leadership | .276 | .032 | .436 | 8.717 | .000 | | | |

Note: Predictors in Step 1: Educational Level; Step 2: Educational Level and Ethical Leadership. Outcome variable: Employee Performance.

**Table 7. The Calculations of Coefficients of The Hierarchical Multiple Regression of Mediating Influence of Internal Motivation on Ethical Leadership and Employee Performance.**

| Model | | Unstandardized Coefficients | | Standardized Coefficients | t | Sig. | R² | Adjusted R² | f² |
|---|---|---|---|---|---|---|---|---|---|
| | | B | Std. Error | Beta | | | | | |
| 1 | (Constant) | 3.987 | .094 | | 42.396 | .000 | | | |
| | Educational Level | −.101 | .048 | −.115 | −2.104 | .036 | 0.013 | 0.010 | 0.013 |
| 2 | (Constant) | 1.953 | .180 | | 10.837 | .000 | | | |
| | Educational Level | −.039 | .040 | −.044 | −.974 | .331 | 0.335 | 0.329 | 0.322 |
| | Internal Motivation | .280 | .034 | .386 | 8.162 | .000 | | | |
| | Ethical Leadership | .204 | .030 | .323 | 6.772 | .000 | | | |

Note: Predictors in Step 1: Educational Level; Step 2: Educational Level, Internal Motivation, and Ethical Leadership. Outcome variable: Employee Performance.

the relationship between ethical leadership and employee performance. As shown in Table 8, ethical leadership had a significant direct effect on internal motivation ($\beta=0.522$, $p<0.001$) and employee performance ($\beta=0.436$, $p<0.001$). Internal motivation also significantly predicted employee performance ($\beta=0.479$, $p<0.001$). Moreover, the indirect effect of ethical leadership on employee performance through internal motivation was significant ($\beta=0.323$, $p<0.001$), confirming partial mediation. Overall, these results support partial mediation among clinical and administrative employees. In conclusion, the analysis showed that internal motivation partially mediates the association between ethical leadership and employee performance among clinical and administrative employees.

To further validate the mediation effect and address potential limitations of the traditional causal steps approach, a bootstrapping mediation analysis was conducted using 5,000 bias-corrected samples in AMOS. The results indicated that all hypothesized paths were statistically significant ($p<.001$), and the 95% bias-corrected confidence intervals did not include zero, confirming the stability of the estimates. Specifically, ethical leadership had a significant positive effect on internal motivation ($\beta=0.257$, 95% CI [0.134, 0.371]), internal motivation significantly influenced employee performance ($\beta=0.281$, 95% CI [0.200, 0.360]), and ethical leadership also had a direct positive effect on employee performance ($\beta=0.208$, 95% CI [0.134, 0.276]). These results verify that internal motivation partially mediates the relationship between ethical leadership and employee performance, supporting the robustness and validity of the proposed model.

## 4. Discussion

The current study sought to analyze the associations between ethical leadership, internal motivation, and employee performance in hospitals in Jordan, as well as the mediating role of internal motivation in this relationship. The study tested four hypotheses, and the results supported all four hypotheses. Our findings demonstrated that ethical leadership and internal motivation are statistically positively and significantly associated with employee performance. Additionally, internal

**Table 8. Model Regression Summary.**

| | | IV | DV | Beta | P |
|---|---|---|---|---|---|
| 1 | The direct effect of ethical leadership on internal motivation | Ethical Leadership | Internal Motivation | .522 | .000 |
| 2 | The direct effect of internal motivation on employee performance | Internal Motivation | Employee Performance | .479 | .000 |
| 3 | The direct effect of ethical leadership on employee performance | Ethical Leadership | Employee Performance | .436 | .000 |
| 4 | The indirect effect of ethical leadership on employee performance | Ethical Leadership | Employee Performance | .323 | .000 |
| | | Internal Motivation | | .386 | |

motivation partially mediated the relationship between ethical leadership and performance, indicating that while ethical leadership directly improves employee outcomes, part of its effect is channeled through fostering employees' intrinsic drive. An important finding of this study is that most demographic characteristics were not significantly associated with internal motivation or employee performance. This suggests that ethical leadership exerts a broadly positive influence across diverse employee groups, regardless of age, gender, or professional background, highlighting leadership practices as a key leverage point for improving performance in healthcare settings.

The findings in this study remained consistent with previous research that underscores the central role of ethical leadership in shaping employees' motivation, performance, and well-being. For instance, in healthcare contexts, [13] demonstrated that ethical leadership not only improved nurses' job performance but also promoted self-compassion, which mediated part of this effect. Similarly, [14] showed that ethical leadership mitigated burnout and enhanced satisfaction and affective commitment among Austrian healthcare professionals, highlighting its importance in sustaining engagement in stressful environments. Our results extend this evidence to the Jordanian healthcare sector, confirming that ethical leadership enhances both motivation and performance within hospitals.

The partial mediation of internal motivation observed in our study is consistent with prior research in both healthcare and non-healthcare contexts. [15] identified motivation as one of the central domains linking leadership behaviors to nursing performance, while [16] found that ethical leadership enhanced organizational citizenship behaviors through trust and psychological well-being. Similarly, [17,18,] emphasized the indirect mechanisms—such as mindfulness and psychological safety—through which ethical leadership impacts performance and accountability. The fact that internal motivation only partially mediated the relationship in our study suggests that other psychological mechanisms, such as trust, self-compassion, or psychological safety, may also play complementary roles in the Jordanian healthcare context.

Beyond healthcare, studies across sectors also confirm the relevance of motivation in explaining how ethical leadership influences employee outcomes. For example, [21,22] demonstrated that internal motivation acted as a critical mechanism through which ethical leadership improved job satisfaction, performance, and innovation. Similarly, [24] reported that internal motivation mediated the link between ethical leadership, commitment, and OCB in the banking sector. Our findings corroborate these international patterns, showing that motivation is a key pathway in Jordanian hospitals as well.

The results also contribute to the broader literature on the role of ethical leadership in fostering prosocial behaviors, organizational identification, and retention. Prior studies [29–31] highlight that ethical leaders build trust, fairness, and psychological safety, which not only reduce turnover intentions but also enhance prosocial motivation and innovative behaviors. Our findings add to this evidence by showing that, in the Jordanian healthcare context, ethical leadership builds internal motivation that strengthens performance outcomes, thus reinforcing the cross-cultural generalizability of ethical leadership's benefits.

Overall, the findings suggest that Jordanian hospitals are effectively adopting ethical leadership practices, as reflected by the relatively high mean scores reported. Nevertheless, the partial mediation indicates that additional organizational factors may further strengthen the impact of ethical leadership. Building on Gupta's (2023) signaling theory perspective [32], it can be argued that ethical leaders in Jordanian hospitals signal fairness, integrity, and support, thereby fostering trust and role-breadth self-efficacy, which could complement internal motivation in improving employee performance. Future research may therefore explore other mediating or moderating mechanisms, such as trust, job satisfaction, or organizational commitment, to provide a more comprehensive understanding of how ethical leadership drives performance in healthcare.

## 5. Study limitations

This study used a convenience sampling approach, which may limit generalizability. Nonetheless, the inclusion of a large and diverse sample may enhance the applicability of the findings. Future research should consider more rigorous sampling methods, such as stratified or random sampling. Although the study revealed a statistically significant association

between ethical leadership, internal motivation, and employee performance, as well as a mediating role of internal motivation, the cross-sectional design limits the ability to draw causal inferences from these relationships. Longitudinal or experimental studies are recommended to better establish causal pathways. This study also was conducted exclusively within public, private, and university hospitals in northern Jordan, which may limit the generalizability of the findings to other regions or healthcare systems. Cultural, organizational, and policy differences may influence the relationships between ethical leadership, internal motivation, and employee performance in different contexts. Therefore, caution should be exercised when applying these results beyond the Jordanian healthcare setting. Future research is encouraged to replicate this study in diverse countries and healthcare environments to enhance the external validity of the findings.

## 6. Conclusion

In sum, the current study moved towards a more comprehensive understanding of the association between ethical leadership and employee performance through internal motivation. Research examining the impact of ethical leadership on employee performance has been extensively studied in research. However, there remains limited research examining the mediating role of internal motivation in the healthcare in Jordan. This study addressed a gap by specifically examining the mediating role of internal motivation on the relationship between ethical leadership and employee performance in clinical and administrative employees working in hospitals in Jordan. Overall, the results of this study were consistent with prior studies in the literature. A significant association was found between ethical leadership, internal motivation, and employee performance.

The findings of the current study can serve as an empirical basis for managers and decision-makers. Specifically, hospital leaders are encouraged to promote ethical leadership behaviors such as fairness, transparency, and ethical decision-making through leadership training programs. In addition, policies that enhance employees' internal motivation, such as recognition systems, autonomy in work roles, and supportive supervision, may contribute to improved employee performance. Integrating ethical leadership principles into hospital management practices may ultimately enhance workforce engagement and quality of patient care. Finally, the study findings offer important insights for future research, specifically research that focuses on healthcare management and policy.

## Supporting information

**S1 File. This is the S1 File Normality Test.**
(DOCX)

**S2 File. This is the S2 File Multicollinearity Test.**
(DOCX)

**S3 File. This is the S3 File Sample Characteristics.**
(DOCX)

**S4 File. This is the S4 File Descriptive analysis of the study variables.**
(DOCX)

**S5 File. This is the S4 File Study Data.**
(CSV)

## Acknowledgments

The authors would like to thank all employees who participated in this study. We are also thankful to the administration of included hospitals and faculty of medicine in Jordan University of Science and Technology.

## Author contributions

**Conceptualization:** Raya Al-Bataineh.

**Funding acquisition:** Raya Al-Bataineh.

**Investigation:** Ameera Hayajneh.

**Methodology:** Raya Al-Bataineh.

**Project administration:** Ameera Hayajneh.

**Writing – original draft:** Ameera Hayajneh.

**Writing – review & editing:** Raya Al-Bataineh.

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
