## [Decision Letter · Decision Letter 0]

25 Jul 2025

Dear Dr. Al-Bataineh,

Thank you for submitting your manuscript to PLOS ONE. After careful consideration, we feel that it has merit but does not fully meet PLOS ONE’s publication criteria as it currently stands. Therefore, we invite you to submit a revised version of the manuscript that addresses the points raised during the review process.

We look forward to receiving your revised manuscript.

Kind regards,

Zahra Masood Bhutta

Academic Editor

PLOS ONE

Journal Requirements:

Reviewers' comments:

Reviewer's Responses to Questions

**Comments to the Author**

1. Is the manuscript technically sound, and do the data support the conclusions?

Reviewer #1: Yes

Reviewer #2: Yes

Reviewer #3: No

2. Has the statistical analysis been performed appropriately and rigorously?

Reviewer #1: Yes

Reviewer #2: Yes

Reviewer #3: No

3. Have the authors made all data underlying the findings in their manuscript fully available?

Reviewer #1: No

Reviewer #2: Yes

Reviewer #3: Yes

4. Is the manuscript presented in an intelligible fashion and written in standard English?

Reviewer #1: No

Reviewer #2: No

Reviewer #3: No

Reviewer #1: Sampling Bias (Page 16): The use of convenience sampling may limit generalizability. Consider acknowledging this limitation and discussing strategies to mitigate bias (e.g., stratified sampling in future studies).

Cross-Sectional Design (Page 10): The design precludes causal inferences. Clarify this limitation and recommend longitudinal studies to establish causality.

Instrument Validation (Page 17–18): While scales like the Ethical Leadership Scale are cited, no evidence is provided for their validity in Jordanian healthcare contexts. Include pilot testing results or cultural adaptation details.

Effect Sizes (Results Section): Report effect sizes (e.g., Cohen’s *d*, R²) alongside *p*-values to assess practical significance (e.g., Table 4–7).

Healthcare-Specific Dynamics (Introduction/Discussion): The manuscript briefly mentions Jordan’s healthcare challenges but does not deeply contextualize how ethical leadership operates in high-stress, resource-constrained hospital environments. Expand using literature on nursing/healthcare leadership.

Mechanisms of Mediation (Discussion): The discussion of how internal motivation mediates the relationship is superficial. Link findings to psychological constructs like psychological safety or organizational identification (see recommended studies below).

Multicollinearity (Page 19): VIF values of 1.00 (Page 19) are unusually low. Verify calculations and ensure multicollinearity diagnostics are accurately reported.

Normality Testing (Page 19): The Kolmogorov-Smirnov test is sensitive to large samples. Supplement with visual checks (e.g., Q-Q plots) in supplementary files.

Data Accessibility (Page 7): The statement “All relevant data are within the manuscript and its Supporting Information files” conflicts with the sample size of 330. Clarify if raw data (e.g., anonymized responses) are included in supplementary files.

Integrate these studies to contextualize ethical leadership and motivation in healthcare:

Linking ethical leadership to nurses’ internal whistleblowing through psychological safety, DOI: 10.1177/09697330241268922

Ethical Leadership, Flourishing, and Extra-Role Behavior Among Nurses, DOI: 10.1177/23779608211062669 as it demonstrates ethical leadership’s impact on nurse outcomes, aligning with your focus on healthcare performance.

Nursing Human Resource Practices and Hospitals’ Performance Excellence: The Mediating Role of Nurses’ Performance, DOI: 10.23750/abm.v92iS2.11247 it supports the mediating role of employee performance in organizational outcomes, reinforcing your theoretical framework.

How Decent Work Influences Internal Whistleblowing Intentions in Nurses: The Parallel Mediating Roles of Felt Obligation and Organisational Identification, DOI: 10.1111/jan.16429

Sparking nurses’ creativity: the roles of ambidextrous leadership and psychological safety, DOI: 10.1186/s12912-024-02277-1

Hypothesis Formatting (Page 15): Use consistent notation (e.g., H₀₁, H₀₂).

Table Clarity (Tables 2–7): Label predictors and outcomes clearly (e.g., “Ethical Leadership → Internal Motivation”).

Grammar: Revise awkward phrasing (e.g., Page 10: “Data was gathered from 330 clinical and administrative employees from February to March 2024 using convenience sampling. from five hospitals...”).

Reviewer #2: 1.Update research articles for updated years

2.Formatting issues

3.Hypothesis table required

4.Give more clarity to results discussion

5.Literature review is missing and support of articles given in discussion and conceptual model draw after results.

Reviewer #3: Critical Review: Major Revisions Required

1.Incorrect Hypothesis Formulation

The paper exclusively presents null hypotheses (e.g., "There is no significant impact of...") rather than alternative hypotheses aligned with the theoretical framework.

Correction: Hypotheses should reflect expected directional relationships. For example:

H₁: "Ethical leadership has a significant positive impact on internal motivation among clinical and administrative employees in northern Jordanian hospitals (public/private/university)."

2.Weak Theoretical Foundation

Each hypothesis lacks grounding in prior empirical evidence.

Action Required: Cite 2–3 key studies per hypothesis to justify expected relationships (e.g., "Consistent with Brown et al. (2005) and Walumbwa et al. (2011), we posit that...").

3.Incomplete Scale Documentation

Scoring methods are unclear (e.g., interpretation of Likert-scale means).

Revisions Needed:

1.Clarify directionality: "Higher item scores indicate stronger employee performance (EP)."

Provide example items (e.g., "Sample EP item: 'I consistently exceed performance targets.'").

Report Composite Reliability (CR > 0.7) and Average Variance Extracted (AVE > 0.5) for validity.

2.Flawed Analytical Sequence

Descriptive statistics (means, SDs, correlations) must precede regression/SEM.

Critical Issue: The null hypotheses cannot support the final model. Use:

Bootstrap mediation analysis (PROCESS/AMOS) with 95% CIs for indirect effects.

SEM fit indices (CFI/RMSEA) if structural paths are tested.

3.Discussion Lacks Hypothesis-Level Analysis

Contrast results with prior work for each hypothesis (e.g., "Contrary to Javed et al. (2017), our findings show...").

4.Explain unexpected outcomes (e.g., low CSR6 loading).

5.Missing "Limitations" Section

Sampling bias (convenience sampling).

Cross-sectional design (causality cannot be inferred).

Context specificity (Jordanian healthcare only).

6.Overall Quality Concerns

The paper suffers from unclear logic (e.g., mismatched hypotheses/model) and requires extensive restructuring.

Recommendation: Study high-quality exemplars (e.g., Academy of Management Journal articles on ethical leadership).

**Do you want your identity to be public for this peer review?** For information about this choice, including consent withdrawal, please see our Privacy Policy

Reviewer #1: No

Reviewer #2: No

Reviewer #3: No

---

## [Author Response · Author response to Decision Letter 1]

21 Oct 2025

Thank you for your feedback and for handling our manuscript. We have carefully reviewed all the comments from the reviewers and editor addressed them point by point. A detailed response table has been uploaded alongside the revised manuscript, outlining our clarifications, changes, and additions for each comment.

---

## [Decision Letter · Decision Letter 1]

23 Dec 2025

Dear Dr. Raya Al-Bataineh,

Thank you for submitting your manuscript to PLOS ONE. After careful consideration, we feel that it has merit but does not fully meet PLOS ONE’s publication criteria as it currently stands. Therefore, we invite you to submit a revised version of the manuscript that addresses the points raised during the review process.

We look forward to receiving your revised manuscript.

Kind regards,

Othman A. Alfuqaha, Ph.D.

Academic Editor

PLOS One

Journal Requirements:

Additional Editor Comments (if provided):

Dear Authors,

After careful consideration of the reviewers’ reports, I am pleased to inform you that the decision on your manuscript is Minor Revision. The reviewers found the study to be of merit; however, several points require clarification and minor improvement. Please revise your manuscript by addressing all reviewers’ comments point by point and provide a clear response explaining how each comment has been handled in the revised version.

In addition, please ensure that the manuscript is thoroughly edited for English language quality by a native English speaker to improve clarity, grammar, and overall readability.

We look forward to receiving your revised manuscript and response to reviewers within the specified revision period.

Kind regards,

Reviewers' comments:

Reviewer's Responses to Questions

**Comments to the Author**

Reviewer #2: All comments have been addressed

Reviewer #4: All comments have been addressed

2. Is the manuscript technically sound, and do the data support the conclusions?

Reviewer #2: Partly

Reviewer #4: Yes

3. Has the statistical analysis been performed appropriately and rigorously?

Reviewer #2: No

Reviewer #4: Yes

4. Have the authors made all data underlying the findings in their manuscript fully available?

Reviewer #2: Yes

Reviewer #4: Yes

5. Is the manuscript presented in an intelligible fashion and written in standard English?

Reviewer #2: Yes

Reviewer #4: Yes

Reviewer #2: Revise Data Analysis chapters: Having errors like regression analysis for demographics and most of variable are insignificant.no discussion regarding this.clarity of results is missing.

Adopt professional reading format for readers interests e.g in sum,authors highlights,it would be passive voice and n Researcher tone like we,you,i words.

Reviewer #4: The authors have addressed the previous reviewer comments very well. The revisions improved the clarity, coherence, and overall quality of the paper.

Minor revision:

1. The manuscript is comprehensive but lengthy especially the Literature Review. Consider reducing repetition and focusing on the most relevant studies to improve readability.

2. The conclusion would benefit from more specific and actionable recommendations for hospital managers and policymakers based on the study findings.

3. Ensure consistency in terminology (e.g., “internal motivation” vs. “intrinsic motivation”) and headings.

**Do you want your identity to be public for this peer review?** For information about this choice, including consent withdrawal, please see our Privacy Policy

Reviewer #2: No

Reviewer #4: No

---

## [Author Response · Author response to Decision Letter 2]

24 Dec 2025

The MEDIATING ROLE OF INTERNAL MOTIVATION ON THE RELATIONSHIP BETWEEN ETHICAL LEADERSHIP AND EMPLOYEE PERFORMANCE IN HOSPITALS IN NORTHERN JORDAN

First of all, we would like to thank the editor and reviewers for their highly efforts and time in reviewing our manuscript. We are highly appreciated your valuable comments and feedback. We just worked hard to address all your comments and we wish that we addressed them effectively.

Reviewers’ Comments

Reviewer #2 Reviewer’s comments Responses

“Revise Data Analysis chapters: Having errors like regression analysis for demographics and most of variable are insignificant.no discussion regarding this.clarity of results is missing.” Done. The Results and discussion sections have been revised. Additional explanatory text has been added to improve clarity and interpretation of the regression results.

Page 13 (Line 390-394)

Page 17 (Line 446)

Page 18 (Line 461-462)

Page 23 (Line 517-520)

“Adopt professional reading format for readers interests e.g in sum,authors highlights,it would be passive voice and n Researcher tone like we,you,i words..”

Done. We have carefully revised the manuscript to adopt a more professional and academic reading format

Reviewer #4 Reviewer’s comments Responses

“The manuscript is comprehensive but lengthy especially the Literature Review. Consider reducing repetition and focusing on the most relevant studies to improve readability.”

Done. We have revised the manuscript to reduce repetition and streamline the Literature Review

Page 4-5 (Line 34-140)

The conclusion would benefit from more specific and actionable recommendations for hospital managers and policymakers based on the study findings Done. We have revised the conclusion to include more specific and actionable recommendations for hospital managers and policymakers, directly derived from the study findings

Page 27 (Line 577-581)

Ensure consistency in terminology (e.g., “internal motivation” vs. “intrinsic motivation”) and headings. We have carefully reviewed the manuscript to ensure consistency in terminology, using “internal motivation” uniformly throughout

---

## [Editor Report · Decision Letter 2]

2 Jan 2026

The MEDIATING ROLE OF INTERNAL MOTIVATION ON THE RELATIONSHIP BETWEEN ETHICAL LEADERSHIP AND EMPLOYEE PERFORMANCE IN HOSPITALS IN NORTHERN JORDAN

PONE-D-24-57949R2

Dear Dr. Raya Al-Bataineh,

We’re pleased to inform you that your manuscript has been judged scientifically suitable for publication and will be formally accepted for publication once it meets all outstanding technical requirements.

Kind regards,

Othman A. Alfuqaha, Ph.D.

Academic Editor

PLOS One

Additional Editor Comments (optional):

Dear Authors,

I am pleased to inform you that your manuscript has been accepted for publication. Congratulations on this achievement.
---

## [Editor Report · Acceptance letter]

PONE-D-24-57949R2

PLOS One

Dear Dr. Al-Bataineh,

I'm pleased to inform you that your manuscript has been deemed suitable for publication in PLOS One. Congratulations! Your manuscript is now being handed over to our production team.

Kind regards,

on behalf of

Dr. Othman A. Alfuqaha

Academic Editor

PLOS One